# How to Modify Drug Release in Paediatric Dosage Forms? Novel Technologies and Modern Approaches with Regard to Children’s Population

**DOI:** 10.3390/ijms20133200

**Published:** 2019-06-29

**Authors:** Monika Trofimiuk, Katarzyna Wasilewska, Katarzyna Winnicka

**Affiliations:** 1Department of Pharmaceutical Technology, Medical University of Bialystok, Mickiewicza 2c, 15-222 Bialystok, Poland; 2Department of Clinical Pharmacy, Medical University of Bialystok, Mickiewicza 2a, 15-222 Bialystok, Poland

**Keywords:** modified release, drug delivery, paediatric formulation development, paediatric dosage forms

## Abstract

In the pharmaceutical technology, paediatric population still presents the greatest challenge in terms of developing flexible and appropriate drug dosage forms. As for many medicines, there is a lack of paediatric dosage forms adequate for a child’s age; it is a prevailing practice to use off label formulations. Children need balanced and personalized treatment, patient-friendly preparations, as well as therapy that facilitates dosing and thus eliminates frequent drug administration, which can be ensured by modified release (MR) forms. MR formulations are commonly used in adult therapy, while rarely available for children. The aim of this article is to elucidate how to modify drug release in paediatric oral dosage forms, discuss the already accessible technologies and to introduce novel approaches of manufacturing with regard to paediatric population.

## 1. Introduction

Creating an appropriate dosage form designed exactly for children still appears as an outgoing challenge for pharmaceutical technology and the distinction between adults and children pharmacokinetics should be considered. The statement that children can be treated as “small adults” is obviously incorrect, particularly in determining the therapeutic doses in individual age groups [1,2,3,4]. In paediatric pharmacotherapy, many factors regarding a convenient dosage form (e.g., age-suitable formulations in proper strength, off label use, and palatability) have to be included. Creating paediatric dosage forms is associated with many difficulties. Therefore, the main goal raised by regulations of European Medicines Agency (EMA) or paediatric scientific network groups is increasing the safety and efficiency of paediatric therapy by the enhancement the quality of clinical studies for children in various age groups (from birth to 18 years old) with better availability of pharmacokinetic data [5,6]. Scientific and governmental initiatives (The Best Pharmaceuticals for Children—BPCA, Paediatric Investigation Plan—PIP) are focused on the development of paediatric dosage forms adjusted to the child’s age and as a consequence of enhancing the efficiency and safety of paediatric therapy [7,8]. The main directives implemented in the appropriate paediatric dosage forms development point basic difficulties connected with paediatric therapy. A special guideline concerning formulation and administration of suitable dosage forms and detailed information on how to use the medicines with regard to children was proposed (State of paediatric medicines in the European Union 10 report) [9]. Due to the special attention focused on the safety of excipients used in the paediatric formulations, the Step and Toxicity of Excipients for Paediatric Patients database has been created (STEP database) [10]. Additionally, EMA gave a support in providing clinical trials with children to promote and harmonize existing paediatric guidelines [7,11,12,13,14,15,16,17,18]. As a result of these acts, the percentage of clinical trials included children (where only one participant was under 18 years at least) has increased from 8.3% in 2007 to 12.4% in 2016. Therefore, it still remains an outgoing problem as there is constantly not enough clinical trials with children compared to the numbers of clinical trials carried out in adults [9]. Challenges associated with conducting clinical trials in children include many childhood diseases, heterogeneity of the population and ethical problems [18]. Main problems related to paediatric therapy are presented in Figure 1.

Traditional dosage forms (e.g., tablets, capsules, and injections) are often not appropriate for children, hence cutting or crushing tablets, splitting capsule and then mixing with the food (solid or liquid) and dilution of injections are common practice [19,20,21,22]. Manipulation with dosage form may cause a risk of damage of formulation structure, hence side effects and changes in pharmacological effect might occur. Despite significant advances in the development of drug dosage forms dedicated especially for children, but unfortunately, unlicensed drugs are still used [1,2,3,4].

Oral route of administration is the most natural way of giving medicines for children. Regardless, child age is still the most popular liquid dosage forms. Their main advantages are safety and ease of swallowing, but the most important and most convenient factor is the possibility of drug administration in a wide group of children by volumetrically measuring a dose, precisely adjusted to a child’s weight or age. Liquid dosage forms are preferred especially for newborns, infants, and smaller children, avoiding the risk of chocking and enhancing the probability of taking a full dose of the drug. For older children (above 6 years), despite liquid formulations, solid dosage forms (tablets, effervescent formulations, orodispersible tablets, films, pellets, or minitablets) could be safely administered. However, the individual swallowing abilities should be considered when administering oral solid forms for children. Nevertheless, numerous studies proved that pre-school (up to 6 years old) and even infants (6 months old) are able to safely swallow particles smaller than 3 mm (minitablets, pellets) [23,24,25,26]. While both liquid and solid forms are available in paediatric therapy, there are not many MR formulations dedicated for children. MR dosage forms ensure drug release in the entire gastrointestinal tract providing constant drug concentration and eliminating the necessity of taking several doses a day, hence improving pharmacotherapy effectiveness [19,27]. Additionally, it is worth emphasizing that utilizing MR technology creates a possibility of increasing drug stability. The most common example is formulating enteric pellets with proton pump inhibitors (omeprazole, pantoprazole), which are unstable in acidic pH. The main reason why MR paediatric forms are lacking includes the clinical aspects like a small range of strength suitable only for particular child’s age. Changing clinical parameters in the chronic diseases requires frequent dose changes depending on the response to therapy, disease management, or age-related changes in strength dose, which requires the availability of a wide range of drug doses on the market. From a technological point of view, the production of various doses is simply unprofitable for pharmaceutical companies, considering that some of the MR technologies are rather expensive (Figure 2) [9,20]. The aim of the current article is to overview available paediatric MR oral liquid and solid formulations depending on the dosage forms and utilized technology, as well as to introduce new approaches and possibilities used in the children’s pharmacotherapy.

## 2. MR Liquid Dosage Forms

Oral liquid formulations (drops, solutions, suspensions, and syrups) are the most popular dosage forms indicated for children of all ages [6]. The main advantage of liquid forms is volumetric dosing, which gives an opportunity to precisely adjust the dose for the specific age groups by measuring an appropriate drug volume. Liquids are easy to swallow and can be administered by child caregivers, without any manipulation before administration, as in the case of tablets (crushing, mixing with food, or fluid). Nevertheless, the limiting factor of using liquids is their physicochemical and microbial instability in an aqueous environment, which entails the need to use preservatives or cosolvents. Additionally, dosing might be a hindering issue in connection with using a not calibrated spoon, oral syringes, or not properly measured drops. Limiting factor of using liquid forms in pediatric preparations is the drug taste as well. Therefore, common practice is to prepare suspensions rather than solutions. For this purpose, excipients such as natural or synthetic flavours or various technologies that minimize the unpleasant feeling of bitterness (e.g., dose sipping technology—straws with coated pellets) can be used [28,29,30,31]. Nonetheless, it is evident that conventional liquids are the most frequently chosen formulations, therefore MR oral liquid forms would be preferable for children to eliminate dosing several times per day, especially in chronic diseases. Not only reducing the frequent drug administration is important, but also providing a favorable pharmacokinetic profile of the drug with keeping drug concentration at a constant therapeutic level [32]. For comparison, in case of MR hard capsules, to achieve a prolonged action of the drug, it is possible to open the MR capsule and to administer the content (e.g., powder, pellets, and minitablets) after mixing with the fluid or food. However, this practice can result in dose errors or MR disturbance if the medication will be chewed not swallowed [19,33]. The following modifications are used for formulating prolonged drug release in liquid dosage forms: Drug/resin complexes, in situ gel formation, microencapsulation, and MR microparticles [34,35]. Despite liquid MR formulations seem to be suitable for applying to children, currently the number of MR liquid forms for children available in the market is limited. Table 1 presents an overview of MR oral liquid formulations with paediatric license and products with finished clinical trials being under the approval registration process by the Food and Drug Administration (FDA).

### 2.1. Drug-Resin Complexes

Utilizing ion-exchange resins for getting a complex with drug is one of the techniques enabling modified drug release in oral liquids (Figure 3) [52]. Nowadays, the most often used resins are cationic exchange-resin with a free sulphuric acid group on the crosslinked polystyrene matrix (extended release utilized in liquids with, e.g., chlorpheniramine and dextromethorphan) and the anionic exchange-resin with amino groups, which are available in the market with paediatric licence (Table 1). Developed complexes are often incorporated into microcapsules (inside the particle or in the coat), lipospheres (lipid microspheres, size 0.01–100 µm), or directly suspended in suspending vehicles. Obtained suspensions are usually administered once (Dyanavel XR^®^, Quilivant XR^®^) or twice daily (Delsym^®,^ MST^®^ Continous^®^) [36,37,39]. Dyanavel XR^®^ utilizes an ion exchange resin, where the amphetamine is bound to the sodium polystyrene sulfate resin through an ionic binding reaction. Dyanavel XR^®^ contains immediate release and extended release components as complexes coated with pH independent polymers: Povidone and polyvinyl acetate [37]. Quillivant XR^®^ is a powder forming an extended-release oral suspension after reconstitution with water. It contains approximately 20% immediate-release and 80% extended-release methylphenidate in drug-polistyrex complex form [39]. MST^®^ Continous^®^ is the example of syrup containing MR granules with morphine sulphate complexed with Dowex 50WX8 cationic exchange resin and suspended in a sugar free medium. It is recommended to take the suspension every 12 h [38].

### 2.2. Microparticles—Spray Drying Technique

The other method of obtaining MR in oral liquids is the spray-drying technique. It creates the possibility of designing microparticles (microspheres, microcapsules) so that drug is incorporated or enclosed in a polymeric shell and then suspended in the liquid (Figure 4) [53,54]. This technique brings the great area on the achieving desired dissolution profile, improved drug stability, and also provides taste masking effect. Depending on the polymers used in the process, various sizes and properties of microparticles might be created. Spray drying is the process by which a dry powder product is formed from the starting solution or suspension [55,56,57,58]. Zmax^®^ is an example of a single-dose, prolonged-release formulation of microspheres for oral suspension containing azithromycin (Table 1). Zmax^®^ was approved as a one-dose-only treatment for mild-to-moderate acute bacterial sinusitis and community-acquired pneumonia. Azithromycin microspheres (50–300 µm) are produced with glyceryl behenate and poloxamer 407 utilization by a combining hot melt extrusion with spray congealing technology. Microspheres were formed by suspending azithromycin in a molten carrier matrix and spraying by a spinning-disk atomizer to form droplets, which congeal into solid microspheres upon cooling [41].

### 2.3. In situ Gel Formation

Modified drug release in liquid formulations could be obtained by the in situ gel formation, which depends on temperature, pH, ions, or UV irradiation. Gel formation from liquid allows achieving a sustained or controlled release profile (Figure 5) [59,60,61]. Physicochemical characteristics like solubility or gelling properties of polymers are crucial for obtaining desired MR. Polymers used for in situ gel formation include gellan gum, xyloglucan, pluronics, tetronics, alginic acid, carbomer, hypromellose, pectins, chitosan, polycaprolactone or poly(DL-lactic acid), or poly(DL-lactide-co-glycolide) [62,63,64]. The in situ forming gel technique is used in SABER^™^ technology (sucrose acetate isobutyrate extended release, SABER^™^ Delivery System), where the biodegradable gel initially appears in low-viscosity fluid form. SABER^™^ systems are dedicated as carriers for drugs in dosage forms administered orally (the clinical trials with SABER^™^ delivery system for bupivacaine). After application, its viscosity increases, making an adhesive gel and MR of drug could be obtained. Orally administered drug diffuses rapidly, leaving in situ drug depot [42,44]. Another system utilizing gel forming matrices is ORADUR™. Gel matrices are able to accumulate high concentrations of drug, with the goal of once and twice-daily dosing. These are also designed to provide controlled long-term treatment preventing the abuse, e.g., the preparations with oxycodone (Remoxy^®^) and methylphenidate [49,50,51]. Unfortunately, formulations based on ORADUR^™^ and SABER^™^ technologies despite completed clinical trials (phase 3) have still not been approved for marketing (Table 1).

## 3. MR Solid Dosage Forms

### 3.1. Matrix and Coated Tablets

Traditional MR tablets are formulated by drug embedding in a hydrophilic (swelling), lipid, or insoluble matrix. A determinant factor for maintaining MR from all tablets types is obtained by incorporation to ensuring a sufficiently long way of drug diffusion. As hydrophilic carriers, water-soluble polymers are utilized: Cellulose derivatives (methylcellulose, hypromellose, and hydroxyprophylmethylcellulose) or sodium alginate. A common feature of these polymers is the formation of highly viscous gel in an aqueous environment hindering the drug diffusion. By suspending drug in the gastrointestinal insoluble carrier, shell tablets are obtained. Among excipients forming insoluble shell inorganic compounds (calcium sulphate, and di- and triphosphate) or organic (ethylcellulose, cellulose acetate) can be distinguished [65,66,67,68]. An interesting example of matrix tablet (intended for children ≥12 years) is Lamictal^®^ XR (Table 2). The tablets are coated with an enteric layer ensuring MR. Simultaneously, there are apertures drilled from the core to the outer layer on both faces of the tablet’s structure (DiffCORE™) to provide a controlled release of drug in the acidic environment of the stomach (Figure 6). Such a combination is designed to control the dissolution rate of lamotrigine over a period of approximately 12 to 15 h, leading to a gradual increase of lamotrigine level in serum [69].

### 3.2. Multiparticulate MR Solid Dosage Forms (MultiP)

Single-unit formulations contain drug in a single tablet or capsule form, whereas MultiP dosage forms comprise of quantity of particles combined into one dosage unit. They may exist as pellets, granules, minitablets, microparticles (microspheres, microcapsules), or nanoparticles with drugs being entrapped in or compacted in the matrix, as well as layered around cores and placed per se in sachets or capsules. MultiP provide many advantages over single-unit systems because of their small size and large surface, which allows leaving the stomach within a short period of time, which results in better distribution and bioavailability improvement. Another advantage of MultiP is the decreased risk of dose dumping due to damaged/broken coating, as well as reduced local irritation as MultiP are more uniformly dispersed in the gastrointestinal tract. Pellets reduce retention in a throat compared to the capsules or powders and improve physicochemical stability. MultiP dosage forms may offer a flexible dosing that allows covering a broad range of doses for different age groups [98,99].

An example of such a system is Multi-Unit Pellet System (MUPS)—utilizing coated pellets for controlled release, usually filled into a capsule or compressed to a tablet form. MUPS technology has been adopted by the pharmaceutical industry as an alternative to conventional immediate or MR tablets. MUPS consisting of pellets ensures divisible dosage form without imparing the drug release characteristic of the individual units. In addition, compared to other carrier systems, MUPS preparations entail a lower risk of irritation and toxicity, dose stability, minimal fluctuations of drug concentration in plasma, and the ability to administer drugs with a narrow therapeutic index. Another advantage is the possibility of using different taste masking techniques. The leading preparation being manufactured utilizing this technology is Losec MUPS with omeprazole for children (approved over 1 year of age and ≥ 10 kg) [85,100]. Formerly, attempts were made to obtain omeprazole as liquid form, however, due to its instability (rapid decomposition at wide range of pH), such a formulation could not be manufactured and launched in the market [101,102].

Another example of MultiP with paediatric licence is Moxatag^™^—prolonged release pulsatile delivery technology (PULSYS^™^) with multiple pellets inside. The typical PULSYS^™^ drug delivery format is a tablet containing pellets with different release profiles. The pellets with amoxicillin are formulated in a proportion that delivers optimal antibiotic levels. Manufacturers assume that this product can be presented in the future as sprinkle granules for the youngest patients [87,88]. Prevacid^®^ is an instance of delayed release capsules with pellets (also available as delayed release orally disintegrating tablet—Prevacid^®^ SoluTab™) containing lansoprazole (Table 2). The pharmacokinetics of lansoprazole was studied in paediatric patients with gastroesophageal reflux disease (GERD) in two separate clinical studies (one children group aged from 1 to 11 years and the second from 12 to 17 years). Each capsule contains enteric-coated pellets consisting of methacrylic acid copolymer providing delayed release of lansoprazole [92,93].

The MR MultiP can also be used in the treatment of hypertension in children above 6 years. The combined (immediate and control) release profile (Coreg CR^®^) or extended release (Toprol-XL^®^) were designed for carvedilol and metoprolol, respectively (Table 2). Coreg CR^®^ is in hard gelatin capsules form filled with immediate or controlled release microparticles. Controlled release of carvedilol is ensured by coating multiparticles using methacrylic acid copolymer [76]. TOPROL-XL^®^ is available as extended release tablets and has been formulated to provide a controlled release of metoprolol for once-daily administration. The tablets comprise a multiple unit system containing metoprolol succinate in a multitude of controlled release pellets. Each pellet acts as a separate drug delivery unit and is designed to deliver metoprolol continuously over the dosage interval [96,103].

The group of patients that requires special attention related to effective pharmacotherapy and has a particular need for taking MR formulations are children suffering from ADHD. MR drug dosage forms allow for extended delivery with less variability throughout the day, improved tolerability and less frequent administration ensuring convenience and adherence, which is important in this patients’ group therapy. Therefore, some pharmaceutical preparations utilizing MR have been introduced for treatment of children with ADHD (Table 2). An interesting example of multiparticulate drug delivery system is SODAS^®^ (Spheroidal Oral Drug Absorption System). Based on the production of controlled release beads, it is possible to provide a number of release profiles, including immediate and sustained release, giving rise to a fast onset of action, which is maintained for 24 h (Figure 7) [104]. Ritalin LA^®^ (methylphenidate hydrochloride) is an extended-release capsule with a bi-modal release profile based on SODAS^®^ system with paediatric license (Table 2). The preparation contains 50% of immediate release beads and 50% extended release beads covered by the polymer overcoat. The first peak in its bimodal profile occurs after 1 to 3 h and the second peak is approximately 6 h post dosage, therefore it is designed to be effective throughout the school day [94]. The second example bases on SODAS^®^ technology is Focalin^®^ XR (Table 2). Similar to the above description, each capsule contains 50% immediate release beads of methylphenidate and 50% extended release beads covered by a polymer, with the difference that the beads can be sprinkled in food [81].

The other example of MR formulation designed for children with ADHD is Adderall^®^ XR administered once daily (Table 2). Each capsule contains a 50:50 ratio of immediate and delayed-release beads. Diffucaps system (utilized in Metadate CD^®^) comprises both immediate release (30%) and extended release (70%) beads. Diffucaps is multiparticulate system, where drug profiles are created by layering a drug onto a neutral core (e.g., sugar spheres, crystals, or granules) followed by the application of a rate-controlling, functional membrane (Figure 8). The physicochemical characteristics of coating materials (being water soluble/insoluble, pH dependent/independent) are conditional on individual drug features. Obtained beads are small in size, approximately 1 mm or less. By incorporating beads with differing drug release properties, combined release profiles can be achieved. Metadate CD^®^ designed to provide efficacy throughout the school day is available in six capsule strengths (with possibility of sprinkle over food) [86].

### 3.3. Minitablets

Minitablets constitute a dosage form that ensures more dose flexibility and ease of drug administration in various children age groups than conventional tablets or capsules. Their main advantage is its small size ranging from 1 to 3 mm, with an average mass from 5 to 25 mg and possibility of adjusting single dose by counting the proper amount of minitablets. Minitablets are produced in the same way as conventional ones, by compression using tableting technology with single or multi-punch. They may appear as individual dosage form or could be delivered in capsules or sachets (Figure 9). The clinical researches have demonstrated that 2 mm tablets can be easily used in six-month old infants and 4 mm in children above one year of age, while orodispersible 2 mm tablets can be administered already for preterm neonates. Moreover, Klingmann et al. proved that children above six-months show greater acceptance of minitablets than syrups. Furthermore, they might provide combined release patterns [25,107,108,109]. Examples of commercially available MR minitablets are: Orfiril Long^®^ and Pancrease MT^®^ (Table 2). Orfiril Long^®^ is provided in hard capsule or single sachet [89]. Pancrease MT^®^ is also provided in minitablets form enclosed inside the capsule [90].

### 3.4. MR Orodispersible Formulations

MR may be also ensured by rapidly disintegrating forms like orodispersible tablets and films. They constitute a relatively new and dynamically developing group of MR formulations. Preparing tablets with such a short disintegration time is a particularly beneficial feature, especially for young patients [110,111,112]. Prevacid^®^ SoluTab^™^ is an instance of delayed release orally disintegrating tablet with compressed MR pellets containing lansoprazole (Table 2). An interesting example of immediate and extended release profiles combined in single dosage form are two commercially available medications namely Adzenys XR^®^-ODT with amphetamine (FDA approval in January 2016) and Cotempla^®^ XR-ODT with methylphenidate group (approval in June 2017). They are indicated for ADHD treatment in children from 6 to 17 years of age. These formulations are available in wide range of doses: Adzenys^®^—3.1 mg, 6.3 mg, 9.4 mg, 12.5 mg, 15.7 mg, 18.8 mg, and Cotempla^®^—8.6 mg, 17.3 mg, and 25.9 mg allowing dosing in a wide age group. These products are obtained by XR-ODT (extended-release orally disintegrating tablet) technology and they dissolve quickly in the mouth (according to the FDA guidelines—up to 30 s or less) so that it can be easily swallowed. The technology utilized in the tablet uses two different types of microparticles: immediately released in 25% (Cotempla^®^) or 50% (Adzenys^®^) and slowly released with the other 75% and 50%, respectively, throughout the day. Two different polymers’ coatings are applied to the XR microparticles: Interior polymer coating as diffusion barrier (ethylcellulose) and exterior polymer coating being pH dependent (methacrylic acid). The technology allows for a drug to be incorporated into orodispersible dosage form using ion resin technology [72,73,77,78,113].

Orodispersible films are defined as thin polymeric films supposed to disintegrate in the oral cavity within seconds (there is no detailed monography in any Pharmacopoeia; FDA indicated 30 s or less as disintegration time). Their size and shape resemble postage stamp, with a thickness ranges from 12 to 100 µm and a surface from 2 cm^2^ to 8 cm^2^ (in the literature, the most frequently encountered dimension is 3 × 2 cm^2^, 2 × 2 cm^2^) [113,114,115]. They are usually manufactured by solvent casting, hot melt extrusion, semisolid casting method, rolling method or electrospinning. There are a number of preparations dedicated specifically for children in this form, but with immediate release, e.g., Pedia-Lax^®^ Quick Dissolve Strip, Orajel^™^ Kids Sore Throat Relief Strips, IvyFilm Kiddies^®^ [116]. In contrast to fast dissolving films, MR release might be obtained by the mucoadhesive effect, which underlies buccal films preparation allowing for prolonged release at the application place. Buccal films are particularly addressed for pre-school and school children since they are thin, adaptable to the mucosal surface and able to offer an exact and flexible dose. Abruzzo et al. have designed buccal films for propranolol hydrochloride (β-blocker used in paediatric patients primarily for the treatment or prevention of cardiac arrhythmias and hypertension) administration. Polymeric layer was prepared by casting and drying of film-forming polymers’ solutions (polyvinylpyrrolidone or polyvinylalcohol with addition of gelatin or chitosan). As a second layer applied onto the primary one in order to obtain prolonged drug delivery and mask its bitter taste, ethylcellulose was utilized. The formulation is intended for children ≥ 2 years of age and body weight around 12 kg [117].

## 4. Excipients Utilized in MR—Safety of Use in Children

The safety of children’s pharmacotherapy depends not only on the drug substance itself, but also on ingredients forming medicines (excipients). The choice of excipients is a crucial factor in the development of medicinal products for paediatric use. Excipient safely and commonly used in adults’ therapy could be harmful for children, e.g., ethanol or propylene glycol cause neurotoxicity; some preservatives like benzyl alcohol, sodium benzoate may lead to allergic reactions. Additionally, a questionable issue is the utilization of parabens in the paediatric population. The most common polymers used for obtaining MR formulations are cellulose derivatives (especially hypromellose and ethylcellulose) utilized as coating polymers, taste masking agents, or e.g. a microparticles matrix (Table 3) [10,68,118,119,120,121,122]. Ethylcellulose as a biocompatible and gastro resistant polymer is used for preparaing sustained release syrup with hydrocodone and chlorpheniramine indicated for children above six-years (Tussionex^®^, Table 1) [40]. Ethylcellulose (in aqueous suspension form) was also used for formulating MR microspheres with mirabegron by the spray drying technique, so oral sustained-release suspensions were obtained [51,52]. Emami et al. used hypromellose for formulating MR suspensions with theophylline [123]. Most cellulose derivatives are generally recognized as safe (GRAS) to use in children pharmacotherapy (Table 3). However, for most excipients used in pediatric formulations (e.g., ethylcellulose acetate, methacrylic acid copolymers, and lauryl sulfate), safety data are still limited. The EMA guidelines serve as a database for assessing the safety profile of excipients, so the presented data must be actualized, related to the age group, and relevant to the maximum daily exposure uptake [10,68,118].

## 5. Novel Technologies 3D Printing for MR Formulations

Pharmaceutical applications of 3D printing have increased over the past years. Printing technologies are cutting edge methods in tablets and films manufacturing. Inject printing is experimentally used for drug printing on different matrices, flexographic printing is employed to coat the drug loaded substrate with a polymeric film. An increasing number of researchers are employing 3D printing technologies to develop oral dosage forms with MR. There is a new approach using a non-contact printing system that incorporates both piezo-electric and solenoid valve-based inkjet printing technologies to deliver both drug and excipients onto the matrix. The main ideas of using this type of technology revolve around the fact that printing technologies would allow to develop pharmaceuticals in a tailored manner to meet some of the envisaged personalization needs of patients for potential use in the paediatric population [124,125]. Recently, 3D printing was utilized to create a multi-active solid dosage form, containing five different drugs within the same capsule, which were autonomously controlled with two separate release profiles, called Polypill^®^ [126]. It would be especially useful for all patients who are taking medicines many times a day. The feasibility of 3D printing coupled with hot melt extrusion to prepare paediatric medicines that can be consumed easily by children from 2–11 years old was introduced. The medicines were designed in such a way to imitate ‘candy-like’ chewable tablets. For the purposes of the study, Starmix^®^ (HARIBO PLC, UK) formulations were printed using indometacine as model drug and hypromellose acetate succinate as the polymeric excipient [127]. The culmination of 3D printing applications in oral dosage forms is the FDA approval (in August 2015) of 3D printed drug product called Spritam^®^ (levetiracetam)—tablets for oral suspension manufactured by using the ZipDose^®^ technology based on a powder bed (liquid 3D printing patented technology). Spritam^®^ became the first 3D-printed drug approved by FDA as a prescription adjunctive therapy for treating patients with epilepsy for children from the age of four-years old (or 20 kg up). The technology enables immediate disintegration of the drug with a sip of water, making it easy for the patients to administer the drug, even in high doses. ZipDose^®^ technology creates a porous formulation using 3D process that binds powders without compression. The method enables delivery of high drug doses of up to 1 g. Drugs formulated using ZipDose^®^ technology are specially designed for people with swallowing difficulties (drug dosage form disintegrate within approximately 11 s according to manufacturer) and those who skip regular drug doses, resulting in ineffective treatment outcomes—the children population perfectly fit in [128,129].

## 6. Conclusions

Pharmacotherapy of children’s population is an important matter in a field of modern pharmaceutical technology. The main problems regarding children treatment result from the diversity of the paediatric population, as well as a relatively small number of appropriate dosage forms, including modern ones. Creating children-made formulation is challenging but an essential task in formulating an appropriate dosage form, which should be adjustable for a wide range of ages, palatable, easy to administer, but first of all safe and effective. Key aspects in modern formulations involve development of novel MR formulations (minitablets, pellets, MR oral liquid formulations) when considering chronic diseases that affects children and minimizing the dose frequency. Simultaneously, safety of excipients and child’s acceptability should be kept in mind. The limited number of MR formulation in the market (especially for children under six-years) arise from the high cost of technologies and lack of relevant clinical trials in the paediatric population. Therefore, new regulations and additional funding opportunities, as well as innovative collaborative research initiatives should be constantly developed.

## Figures and Tables

**Figure 1 ijms-20-03200-f001:**
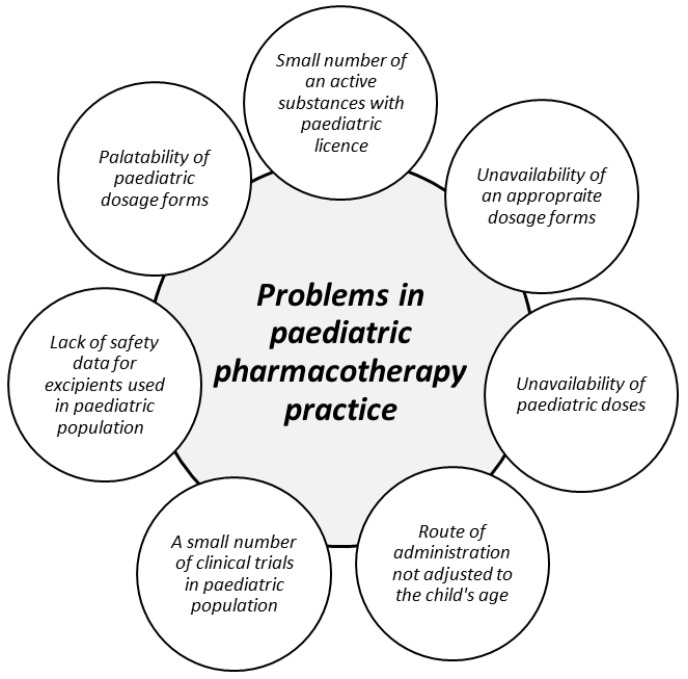
Problems in paediatric pharmacotherapy practice [11,19,20].

**Figure 2 ijms-20-03200-f002:**
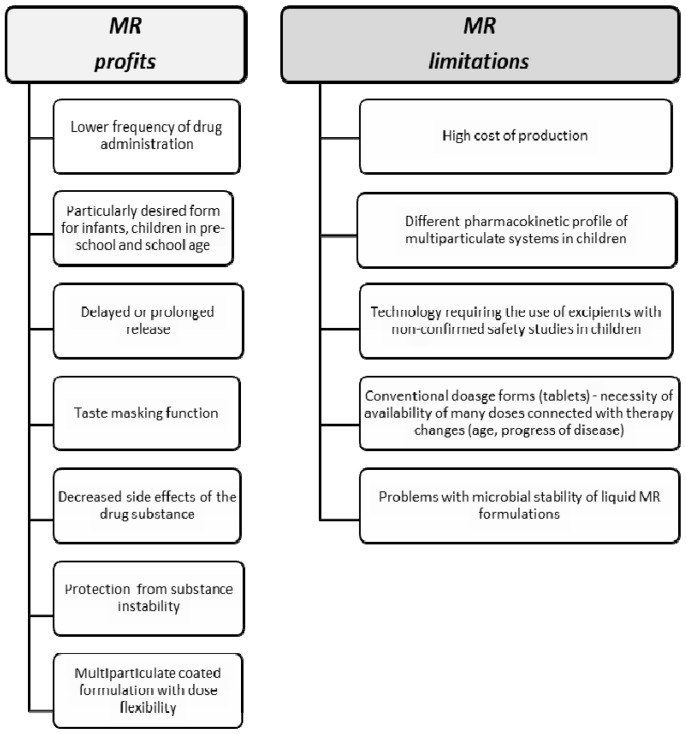
Profits and limitations of use the modified release (MR) formulations in paediatric population.

**Figure 3 ijms-20-03200-f003:**
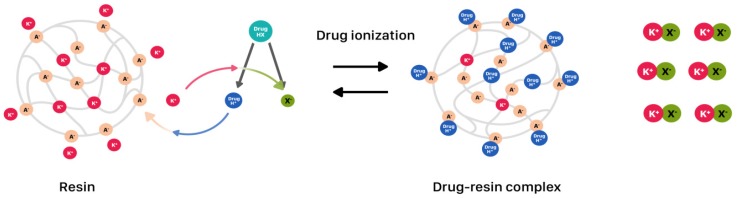
Drug release modification utilizing drug-resin complexes.

**Figure 4 ijms-20-03200-f004:**
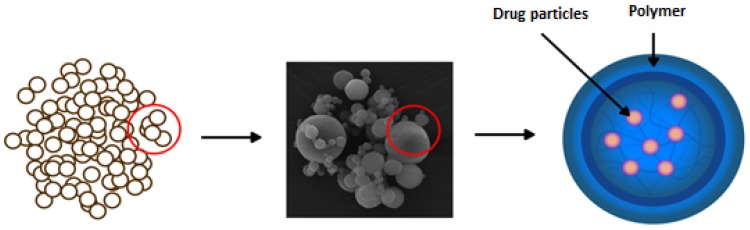
Drug release modification using spray-dried microparticles.

**Figure 5 ijms-20-03200-f005:**
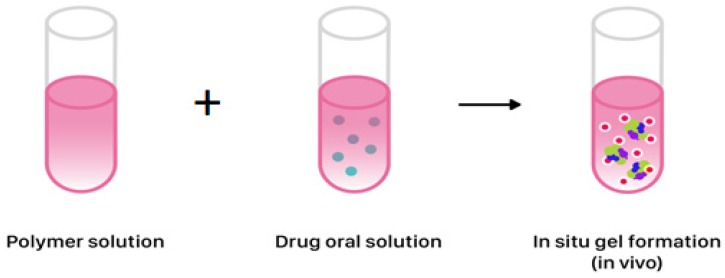
In situ gel formation from liquid for obtaining sustained or controlled release profile of drug in liquid dosage form.

**Figure 6 ijms-20-03200-f006:**
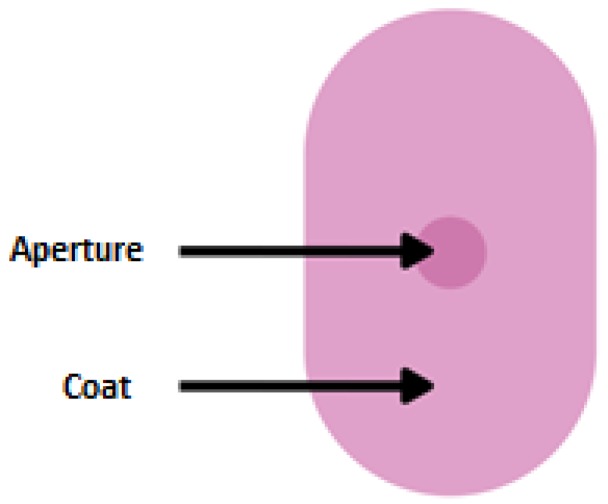
Scheme illustration of DiffCORE^™^ system.

**Figure 7 ijms-20-03200-f007:**
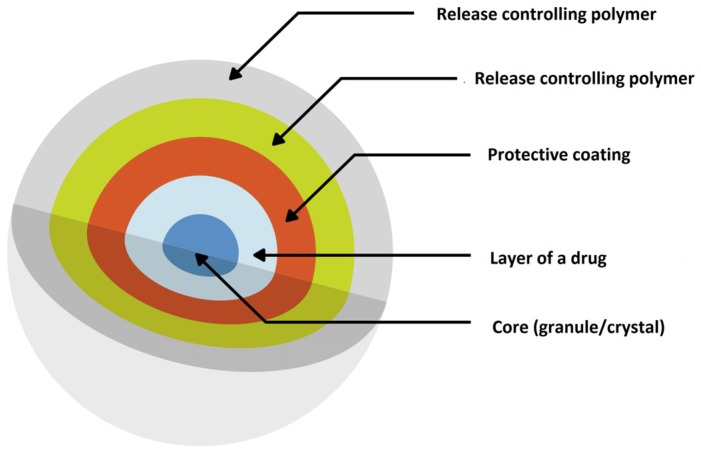
Scheme illustration of SODAS^®^ delivery system modified from Elan drug technologies [105].

**Figure 8 ijms-20-03200-f008:**
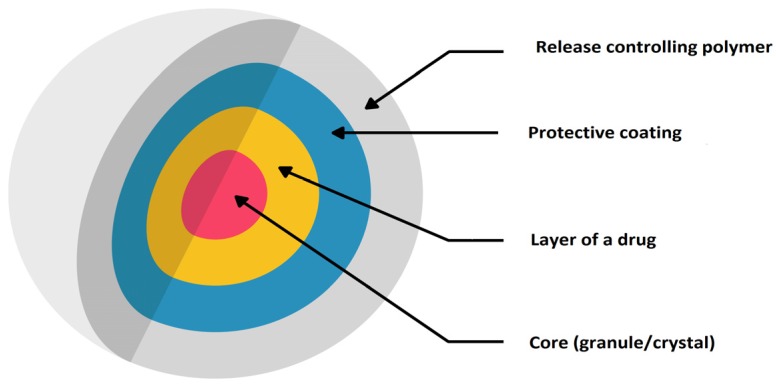
Scheme illustration of Diffucaps^®^ bead technology modified from Weil [106].

**Figure 9 ijms-20-03200-f009:**
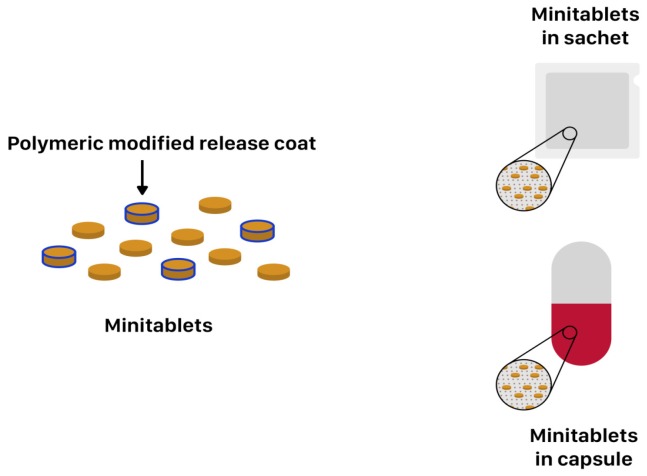
Scheme illustration of minitablets placed in capsule and sachet.

**Table 1 ijms-20-03200-t001:** Modified release oral liquid dosage forms—an overview of available formulations with paediatric license and drugs with finished clinical trials being under approval registration process by the Food and Drug Administration (FDA).

Product (Manufacturer)	Drug	Dosage Form	Polymer (MR Technique)	Paediatric Licence	Indication	References
Delsym^®^ (Reckitt Benckiser LLC.)	dextromethorphan	extended-release suspension	ion exchange resin *drug/polymer complexation*	≥4 year	Cough	[36]
Dyanavel XR (TrisPharma)	amphetamine	extended-release suspension	sodium polystyrene sulfonate *drug/polymer complexation*	≥6 years	attention-deficit hyperactivity disorder (ADHD)	[37]
MST^®^ Continus^®^ (Napp Pharmaceuticals Limited)	morphine	prolonged-release suspension	cationinc exchange resin *drug/polymer complexation*	≥1 year	Pain	[38]
Quillivant XR (Pfizer)	methylphenidate	extended-release suspension	sodium polystyrene sulfonate *drug/polymer complexation*	≥6 years	ADHD	[39]
Tussionex^®^(UCB)	Hydrocodone chlorpheniramine	extended-release suspension	ion exchange resin *drug/polymer complexation microparticles*	≥6 years	common cold, flu	[40]
Zmax^®^ (Pfizer)	azithromycin	extended-release suspension	glyceryl behenate and poloxamer 407 *microspheres*	≥6 month	bacterial infections	[41]
under approval registration process by FDA	bupivacaine	NA *	SABER^™^ Delivery System in situ *gel formation*	NA *	local anesthetic	[42,43,44]
under approval registration process by FDA	methylphenidate	NA *	ORADUR^™^ in situ *gel formation*	NA *	pain	[45,46]
under approval registration process by FDA	mirabegron	sustained-release suspensions	microspheres with lauryl sulfate salt/complex	NA *	urinary incontinence	[47,48]
under approval registration process by FDA (Remoxy^®^)	oxycodone	NA *	ORADUR^™^ in situ *gel formation*	NA *	Pain	[49,50,51]

* NA: no data available.

**Table 2 ijms-20-03200-t002:** Modified release solid dosage forms—an overview of available formulations with paediatric licence.

Product (Manufacturer)	Drug	Dosage Form	Polymer/MR Technique	Paediatric Licence	Indication	**References**
Aciphex^®^ Sprinkle^™^ (Eisai Management Co., Ltd.)	rabeprazole	capsules with granules	ethylcellulose *(delayed release)*	≥1 year	reflux	[70]
Adderall XR^®^ (Shire US Inc)	mixed salts of single-entity amphetamine product	capsules with granules	hypromellose, methacrylic acid copolymer *(immediate and delayed release)*	≥6 years	attention deficit hyperactivity disorder (ADHD)	[71]
Adzenys XR-ODT^™^ (Neos Therapeutics)	amphetamine	extended release orally disintegrating tablets (ODT)	methacrylic acid, ethylcellulose *(ion resin technology*)	≥6 years	ADHD	[72,73]
Azulfidine^®^ (Pfizer)	sulfasalazine	tablets	cellulose acetate phthalate (*delayed release*)	≥6 years	mild to moderate ulcerative colitis Crohn disease – off labell use	[74]
Concerta^®^ (Janssen-Cilag)	methylphenidate	tablets	cellulose acetate (*matrix*, *extended release*)	≥6 years	ADHD	[75]
Coreg CR^®^ (GSK)	carvedilol	capsules	methacrylic acid copolymers (*controled release*)	≥6 years	hypertension	[76]
Cotempla XR-ODT^®^ (Neos Therapeutics)	methylphenidate	orodispersible tablets	methacrylic acid, ethylcellulose, polystyrene sulfate (*immediate and extended release*)	≥6 years	ADHD	[77,78]
Creon^®^ (Solvay Pharmaceuticals)	pancreatic enzymes	Minitablets (in capsules)	dibuthyl phthalate, hypromellose phthalate (*delayed-release*)	≥6 years	chronic pancreatitis, cystic fibrosis	[79]
Finlepsin^®^ (Teva Pharmaceuticals)	carbamazepine	tablets	Eudragit RS 30D, Eudragit L 30 D (*extended release*)	≥6 years	epilepsy	[80]
Focalin^™^ XR (Elan Holdings Inc.)	dexmethylphenidate	capsules	ammonio methacrylate copolymer *(extended release)*	≥6 years	ADHD	[81]
GranuPAS^®^ (Lucane Pharma)	para-aminosalicylic acid	gastro-resistant granules in sachet	methacrylic acid – ethyl acrylate copolymer (1:1) *(extended release)*	≥1 year	tuberculosis	[82]
Kapvay ^®^ (Concordia Pharmaceuticals Inc.)	clonidine	tablets	hypromellose (diffusion from gel matrix structure -*extended release*)	≥6 years	ADHD	[83]
Keppra XR^®^ (UCB)	levetiracetam	tablets	hypromellose, polyvinyl alcohol- (diffusion from gel matrix structure -*extended release*)	≥12 years	epilepsy	[84]
Lamictal^®^ XR (GSK)	lamotrigine	tablets	methacrylic acid copolymers (*extended release)*	≥12 years	epilepsy	[69]
Losec MUPS (AstraZeneca AB)	omeprazole	gastro-resistant tablets with coated pellets	methacrylic acid – ethyl acrylate copolymer (1:1) dispersion *(delayed-release)*	≥1 year	reflux	[85]
Metadate CD^®^ (UCB Manufacturing, Inc.)	methylphenidate	capsules with granules	hypromellose, polyethylene glycol, ethylcellulose *(immediate and extended release)*	≥6 years	ADHD	[86]
Moxatag^™^ (MiddleBrook Pharmaceuticals, Inc.)	amoxicilin	tablets	prolonged-release pulsatile delivery technology MUPS	≥12 years	tonsillitis, pharyngitis	[87,88]
Orfiril Long (Desitin Arzneimittel GmbH)	natrii valproas	Minitablets *(in sachet or capsule*)	ethylcellulose, ammonium methacrylate copolymer *(extended release)*	≥6 years	epilepsy	[89]
Pancrease MT^®^ (McNeil)	pancreatic enzymes	enteric-coated minitablets in capsule	methacrylic acid ethyl acrylate copolymers (*delayed release*)	from birth	chronic pancreatitis, cystic fibrosis	[90]
Pentasa^®^ (Ferring GmbH)	mesalazine	granules	Ethylcellulose *(prolonged release)*	≥6 years	Crohn’s disease	[91]
Prevacid^®^ SoluTab^™^ (Takeda)	lansoprazole	orodispersible tablets	methacrylic acid copolymer *(delayed relese)*	≥1 year	reflux	[92,93]
Ritalin^®^ LA (Novartis)	methylphenidate	capsules	ammonio methacrylate copolymer, gelatin, methacrylic acid copolymer *(delayed relese)*	≤ 6 years	ADHD	[94]
Tegretol^®^ XL (Novartis)	carbamazepine	tablets	ethylcellulose dispersion (*matrix*, *extended release*)	≤ 6 years	epilepsy	[95]
TOPROL-XL^®^ (Aralez Pharmaceuticals)	metoprolol	tablets	cellulose compounds (*extended release*)	≥6 years	hypertension	[96]
Viramune^®^ (Boehringer Ingelheim International GmbH)	nevirapine	tablets	hypromellose *(prolonged release)*	≥3 years	human immunodeficieny virus (HIV) infection	[97]

**Table 3 ijms-20-03200-t003:** Safety data of excipients used in paediatric modified release preparation [10,68].

Excipient		Paediatric Safety Data Use	Main Function in Formulation
Cellulos derivatives	cellulose acetate	NA *	MR
	cellulose acetate phthalate	NA *	MR
	carmellose sodium	yes	suspending agent
	ethylcellulose	yes	MR taste masking
	hypromellose	yes	suspending agent MR taste masking
	methylcellulose	yes	suspending agent
ion exchange resin		NA *	drug/polymer complexation
methacrylic acid copolymers		NA *	MR
sodium polystyrene sulfate		NA *	drug/polymer complexation MR
sodium alginate		NA *	MR
calcium sulfate		NA *	MR
lauryl sulfate		NA *	MR
polyvinyl alcohol		NA *	MR

* NA: no data available.

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
