# Peer review of "How to Modify Drug Release in Paediatric Dosage Forms? Novel Technologies and Modern Approaches with Regard to Children’s Population"

_ijms, 2019, doi:10.3390/ijms20133200_

Round 1
Reviewer 1 Report
The review entitled How to modify drug release in paediatric dosage forms? Novel technologies and modern approaches with regard to children’s population, written by the authors Trofimiuk M at al. is interesting and well conceptuated, nevertheless, there are some issues that must be addressed before publishing:
Line 78: in addition to oral dosage forms (DF) rectal should also be discussed. On my personal experience as a mother of two, suppositories are very/most convenient dosage form in very small babies (e.g. applying antipyretic during the night to 0 - 1,5 or 2 year old babies).
Line 120: liquid DF are problematic also for highly potent APIs (with poor aqueous solubility), for those with unpleasant taste…that cannot be formulated as suspension.
Line 159: what are the lipospheres (the size… )?
Lines 174: spray drying section; to my knowledge Zmax are not produced by spray drying technology but by combining hot melt extrusion technology with spray congealing technology (which is indeed performed in spray drying device). Nevertheless, spray congealing does not equal to spray drying.
Line 203: what is the gelling mechanism upon oral administration of SABER? Is it known? It would be interesting to explain/present it. The same goes for ORADUR and Remoxy.
Lines 237-239: some information is missing. How can DiffCORE™ system control the drug release for 12-15h, when the orifice is drilled only in the enteric coating? Is the tablet meant to retain within the stomach for the whole time? If so, how is this obtained? Please clarify a bit more into details.
Line 261: Another advantage of MultiP is decreased risk of dose dumping due to damaged/broken (enteric) coating as well as reduced local irritation due to distribution of MultiP throughout the GIT.
Line 335: Minitablets could also be produced by hot melt extrusion technology.
Lines 352 – 362: it would be useful to list the exact (or at least approximate) disintegration time; e.g. how fast is “quickly”.
Lines 365-366: I believe it is just the opposite as written – it would be more logically that ethylcellulose is the polymer responsible for presenting the diffusion barrier and methacrylic acid for the pH-dependent polymer coating (and not the vice-versa as written in the manuscript)? If not, than some other vital excipients must be added to the ethylcelulose layer that must be presented in the text.
Line 362: what is the approx. size of these oral films?
Line 404: I believe that “lauryl sulphate” is not a cellulose derivative as listed.
Line 387- section on excipients safety: this is extremely important topic, and as such it could be extended a bit, e.g. the authors could also discuss shortly the (un)safety of polysorbates and PEGs (toxic when administered parenteral).
General: what about the dose sipping technology (the straws (pre)filled with fast dispersing granulate)? I believe they could also be mentioned in the review.
As written above, the review is interesting; however some issues must be corrected/ presented more into details.
Author Response
Response to the Reviewer 1
1. We agree that rectal administration is convenient route in small babies, but the goal of our work was to review oral liquid and solid dosage forms with modified release regarding to children population, so suppositories do not concern the article’s topic.
2. We agree with the Reviewer that API solubility and API taste masking is a problematic issue concerning liquid dosage forms. Therefore, the manuscript has been supplemented as follow (lines 122-125): “Therefore, common practice is to prepare suspensions rather than solutions. For this purpose, excipients such as natural or synthetic flavours or various technologies that minimize the unpleasant feeling of bitterness (e.g. dose sipping technology - straws with coated pellets) can be used”.
3. According to the Reviewer suggestion, we added information about lipospheres size as follow (line 163): “Developed complexes are often incorporated into microcapsules (inside the particle or in the coat), lipospheres (lipid microspheres, size 0.01 – 100 µm) or directly suspended in suspending vehicles.”
4. We agree with the Reviewer’s statement that Zmax® is not produced straight by spray drying technology. According to the suggestion of the Reviewer, the sentence was improved as follow (lines 189-193): “Azithromycin microspheres (50-300 µm) are produced with glyceryl behenate and poloxamer 407 utilization by a combining hot melt extrusion with spray congealing technology. Microspheres were formed by suspending azithromycin in a molten carrier matrix and spraying by a spinning-disk atomizer to form droplets, which congeal into solid microspheres upon cooling.”
5. We agree that it would be interesting to describe in details Saber™ and Oradur™ gelling mechanism, but the entire mechanism and composition of the formulations are protected by patent. It was described according to the available literature.
6. Taking into account the comment raised by the Reviewer, the information concerning DiffCORE™ has been presented in a more precise way. The sentence was improved as follow (lines 238-241): “Simultaneously, there are apertures drilled from the core to the outer layer on both faces of the tablet’s structure (DiffCORE™) to provide a controlled release of drug in the acidic environment of the stomach (Figure 6).”
7. According to the Reviewer’s suggestion we supplemented the text as follow (lines 262-264): “Another advantage of MultiP is a decreased risk of dose dumping due to damaged/broken coating as well as reduced local irritation as MultiP are more uniformly dispersed in the gastrointestinal tract.”
8. In our manuscript we have mentioned the main, most popular and economic method of obtaining minitablets, without going into details of additional technologies. We just wanted to outline the subject.
9. Taking into account the comments raised by the Reviewer, we supplemented the manuscript as follow (line 368): “Tablets disintegrate according to the FDA guidelines - up to 30 sec or less (...).”
10. We agree that the polymers were mistaken and should be put inversely as follow (lines 371-373): “Two different polymers’ coatings are applied to the XR microparticles: interior polymer coating as diffusion barrier (ethylcellulose) and exterior polymer coating being pH dependent (methacrylic acid).”
11. The text has been supplemented with the following information in accordance to the Reviewer (lines 377-379): “Their size and shape resemble postage stamp, with a thickness ranges from 12 to 100 µm and a surface from 2 cm2 to 8 cm2 (in the literature, the most frequently encountered dimension is 3x2 cm2, 2x2 cm2) [117-119].”
12. According to the Reviewer suggestion we changed the sentence as follow (lines 411-413): “However, for most excipients used in pediatric formulations (e.g. ethylcellulose acetate, methacrylic acid copolymers, lauryl sulfate), safety data are still limited.”
13. The issue of our work was to review oral dosage forms with modified release, so excipients used for parenteral formulations do not concern the article’s topic. \
14. Dose sipping technology was mentioned in the lines 122-125: “Therefore, common practice is to prepare suspensions rather than solutions. For this purpose, excipients such as natural or synthetic flavours or various technologies that minimize the unpleasant feeling of bitterness (e.g. dose sipping technology - straws with coated pellets) can be used”.
We thank the Reviewer for the revision and insightful suggestions.
Reviewer 2 Report
In the review manuscript, the authors illustrated “Paediatric dosage” for its properties and applications. The manuscript was well planned and structured and thus, it is well suited for this publication without any further change.
Author Response
We thank the Reviewer for the review and acceptance of our manuscri[t.